# The Linkage of Digital Transformation and Tourism Development Policies in Indonesia from 1879–2022: Trends and Implications for the Future

Windi Dwi Nanda *[ID], Ida Widianingsih *[ID] and Ahmad Zaini Miftah [ID]

Public Administration Department, Faculty of Social and Political Sciences, Universitas Padjadjaran, Bandung 40135, Indonesia; a.z.miftah@unpad.ac.id
* Correspondence: windi19002@mail.unpad.ac.id (W.D.N.); ida.widianingsih@unpad.ac.id (I.W.)

**Abstract:** This research analyzes digital transformation and tourism development in Indonesia based on policies issued by the central government. The policy was issued in the period 1879–2022, or, precisely, during the Dutch colonial period in Indonesia until the COVID-19 pandemic. This study aims to analyze digital transformation policies and tourism policies in Indonesia that are historically linked, as well as their implications for the future. We analyze the trends and times of the COVID-19 pandemic and their implications for the future. The method used in this research is content analysis by analyzing policy texts quantitatively (number of policies in a certain time) and qualitatively (topics and content). The dataset obtained for analysis in this research contains 87 policies on digital transformation and tourism development with various forms of policies. This study found a linkage between digital transformation policies and tourism development, especially tourism development policies toward digital transformation. In addition, during the COVID-19 period, both policies reached the highest number compared to previous years. This allows for more supportive policies to be born in the coming years and implies opportunities for establishing policies on the use of technology in tourism management.

**Keywords:** digital transformation; tourism policy; content analysis; policy; COVID-19

## 1. Introduction

Sustainable development is essential with regards to considering the imbalance between the 'two worlds' of natural and artificial—a problem today [1,2]. Sustainable development on a regional scale requires an understanding of the dynamics and the transformation of local development to maintain superior regional resources [3]. Local, sustainable development includes the transformation of economic, social, and environmental dimensions in a region, as well as the relationship between the public, private sector, educational institutions, government, and Non-Governmental Organizations (NGOs) [4]. Subsequently, the United Nations and member countries agreed on 17 Sustainable Development Goals (SDGs) [1,5]. In Indonesia, SDGs are supported through Presidential Regulation Number 111 of 2022 concerning the implementation of Sustainable Goal Achievement. Based on the Sustainable Development Report [6], Indonesia occupies the 82nd position out of 163 countries in the world, with 69.2 points.

SDGs are supported by digital transformation in capturing patterns and trends of applicable information [7]. Digital transformation is a response to the demands of rapid administration by the public, thus motivating related parties to change [8]. In the last two decades, the development of Information and Communication Technology (ICT) has encouraged various sectors to adopt business processes, service delivery, relationships, and Human Resources (HR) development, which affects other aspects of the organization [9].

The COVID-19 pandemic has caused tragedy for many people regarding health, the economy, and society, which can be seen in changes in social dynamics and interactions [10].

One of the efforts made by the government to prevent the spread of COVID-19 was to limit the movement of people. This COVID-19 measure displays inclusive governance but creates trade-off consequences for public health and the economic system, making it a cross-sectoral issue [11,12]. This outbreak also affects and stops the achievement of SDGs for member countries [11,13]. With this impact, it is necessary to analyze the capacity of each policy actor in the decision-making process [14]. The COVID-19 pandemic has infected more than 180 million of the world's population, with 4 million deaths that have spread to more than 200 countries [15].

Tourism is the worst affected area by the COVID-19 pandemic due to it depending on other fields and being vulnerable to sudden events, such as natural disasters, terrorist attacks, and pandemics [10,16]. In 2020 and 2021, world tourism revenues fell by more than USD 4 trillion, and the unemployment rate increased by 5.5–15% [15]. In Indonesia, foreign tourists decreased by 76.8%; there were only 3.77 million tourists in 2020, while in the previous year, 2019, there were 16.1 million visits [17].

With the COVID-19 pandemic, the economic system is forced to adapt faster and remain productive. Hence, public services that utilize digital technology to reduce risks that may be more important during the COVID-19 period [18]. In addition, it is necessary to change the policy on service standards during COVID-19, which is essential so that people can still access services [19]. Various parties, such as communities, institutions, and governments, also collaborate in overcoming daily problems by using technology to improve the quality of life of the group; this is then referred to as a smart community, with components of connectedness, infrastructure, and a sense of ownership [20,21]. Shifting activities of previous face-to-face meetings to virtual ones indirectly supports digital transformation during the COVID-19 period [22].

Tourism, one of the sectors that apply technology with the rapid use of digital platforms, has changed practices widely and profoundly [23]. The tourism sector is one of profitable economic growth, and, in the past 10 years, tourism has become the largest category of Internet service sales globally due to support by digitalization [24,25]. Based on the 2022 State Budget Financial Memorandum (APBN), the tourism sector contributed 4.2% of Indonesia's Gross Domestic Product (GDP) in 2021. The use of technology in everyday life can make it easier for people to carry out activities, such as those carried out by Tarumajaya Village, West Java Province, Indonesia. However, there are obstacles, namely, the lack of access to technology for village officials and the local community to increase village capacity in developing local potential, such as tourism [26]. Tarumajaya Village has a community institution called the Tarumajaya Smart Community Information Group (KIM), which is responsible for forwarding information related to the village to local and outside communities. This information is conveyed through social media, such as Instagram, Youtube channels, and Facebook. KIM Cerdas Tarumajaya's Instagram account, kimcerdastarumajaya, has 353 uploads and 976 followers. Meanwhile, a Youtube channel in service since 12 December 2019, called KIM Cerdas Tarumajaya, with a total of 74 videos, has been viewed 12,683 times with 433 subscribers.

Research by Rahman, Hassan, and Sifa [10] found that the COVID-19 pandemic has had a substantial impact on tourism events and the international tourism market as a result of perceptions of risk due to uncertain conditions during a pandemic. Tourism organizers are accelerating technology by redesigning innovative travel experiences to remain adaptable amid the COVID-19 pandemic. Furthermore, another study by Srisawat Zhang, Sukpatch, and Wichitphongsa [27] researched foreign tourists traveling to Thailand. This study revealed that accommodation and information significantly influence travel decisions during the COVID-19 pandemic. This study provides recommendations for post-COVID-19 tourism policies to provide room for improving the quality of accommodation and the availability of information for tourists.

With regard to policy changes, the growth of the contents and issues of the regulations that have been determined have three aspects to be analyzed: the period, the level of policy implementation, and the institution/organization that issued the policy [28]. The

use of technology in tourism is an important aspect that can increase the competitive advantage in the promotion of tourism and strengthen strategies in tourism operations, so exploration is needed in the field of research at the intersection of tourism studies and technology [29,30]. Furthermore, this utilization requires a policy foundation to harmonize the use of technology and tourism development that impacts sustainable development with the support of collaboration from various actors to develop policies [31].

Based on this explanation, studies on digital transformation policies and tourism development are needed, especially for the alignment of the two fields that affect sustainable development. Therefore, the purpose of this study is to analyze digital transformation policies and tourism policies in Indonesia, specifically how they are historically linked, as well as their implications in the future. The following research question was proposed: "How do the digital transformation policy and the tourism policy in Indonesia interlock historically, and what are their implications in the future?"

The study consists of five parts. The first part is an introduction that explains the conceptual and contextual phenomena raised. The second part is about reviews, based on the literature, related to digital transformation, tourism, and sustainable tourism. The third section discusses methods and information for obtaining data. The fourth section presents the results of the research findings along with their descriptions. The fifth section contains discussions, implications, limitations, and future research directions based on the interpretation of the results section.

## 2. Theoretical Background

Digital transformation is defined as an organization's response to environmental change as a combination of technological and technical resources and involves changes to processes and culture to increase efficiency by automating work [32,33]. Digital transformation is identified by complexity, speed of change, and uncertainty [34]. Digital transformation has four essential elements: the unit where digital transformation takes place, the scope of transformation, instruments, and outcomes [33]. Digital transformation is generally capable of having a positive impact on economic development, employment, supply chain efficiency, geopolitics, access to digital services, and cost efficiency [35,36]

There are two perspectives when looking at digital transformation, namely, the technical perspective, which focuses on the role of information technology in business processes, and the value perspective, which states that digital transformation at the level of thinking changes the value proposition and operating model in the use of technology [37]. The use of technology creates significant challenges but becomes a benchmark for organizational development and a competitiveness booster for organizations [35]. Every country today faces the pressure of digital transformation regarding the products and services produced and how quickly they can change [38]. Technology has a significant and essential impact on society. Technology should be developed, used, and promoted to support the welfare of society. Because, when there is a failure to consider the social dimension of technology, there will be a negative impact [39].

Digital transformation is a call that indicates that the use of technology is essential, including its policies [40]. Policies can guide all stakeholders in coordinating strategies to achieve common goals [31]. Digital transformation policy is mainly mixed with other fields, and its goals tend to be ambiguous. Additionally, in the context of its impact on society, it can be seen as a Pandora's box with unknown consequences [38]. However, some countries have had specific policies regarding digital transformation in the past [31].

Governments and digital innovation organizations obtain references through digital transformation policies to formulate effective and sustainable transformation policies [41]. Standardization of products, processes, and governance systems is essential for continuous transformation, reflecting the need to increase human life expectancy [42]. Based on Gomez-Trujillo and Gonzalez-Perez [43], transformation policies are initiated by the government, for example, in India through the "Digital India Programme", which aims to provide a digital platform and carry out an electronification processes and increase transparency

that can reduce the level of corruption so that it is in line with one of the objectives of the SGDs. Therefore, by learning technology skills and data interpretation, we can improve sustainability for specific fields and professions [44].

Technological developments are revolutionizing tourism and forcing it to change, especially with the nature of its network that unites service providers and the community as users [34]. To increase the effectiveness of tourism, it can take advantage of ICT developments that provide new experiences for tourists [45]. Policy planning policies should focus on increasing tourism resilience in the post-pandemic period with sustainable practices and minimizing negative impacts on local communities and environments [27].

The COVID-19 pandemic impacts sustainable tourism trends and digital technology by implementing environmentally friendly and socially responsible practices [27]. Sustainable tourism reflects the implementation of sustainable development, which more broadly has a dominant position in tourism studies and tourism policy and planning processes [46]. Sustainable tourism policy involves stakeholders from different sectors as a key factor [15]. Sustainable tourism policies can also benefit the community and align with sustainable goals, as well as help the recovery of the tourism industry after the COVID-19 pandemic [27].

## 3. Materials and Methods

This research uses a content analysis approach with the identification and interpretation of policy texts to analyze policies quantitatively (number and frequency of policies) and qualitatively (themes, objectives, and policy patterns) [47,48]. The following data show information concerning the process of collecting data in this research (see Figure 1). The data collection and analysis process consists of several stages, referring to Hall and Steiner [47]. The following steps of data collection and analysis are adjusted according to the research objectives (see Figure 2).

**Figure 1.** Information in the process of collecting data in this research.

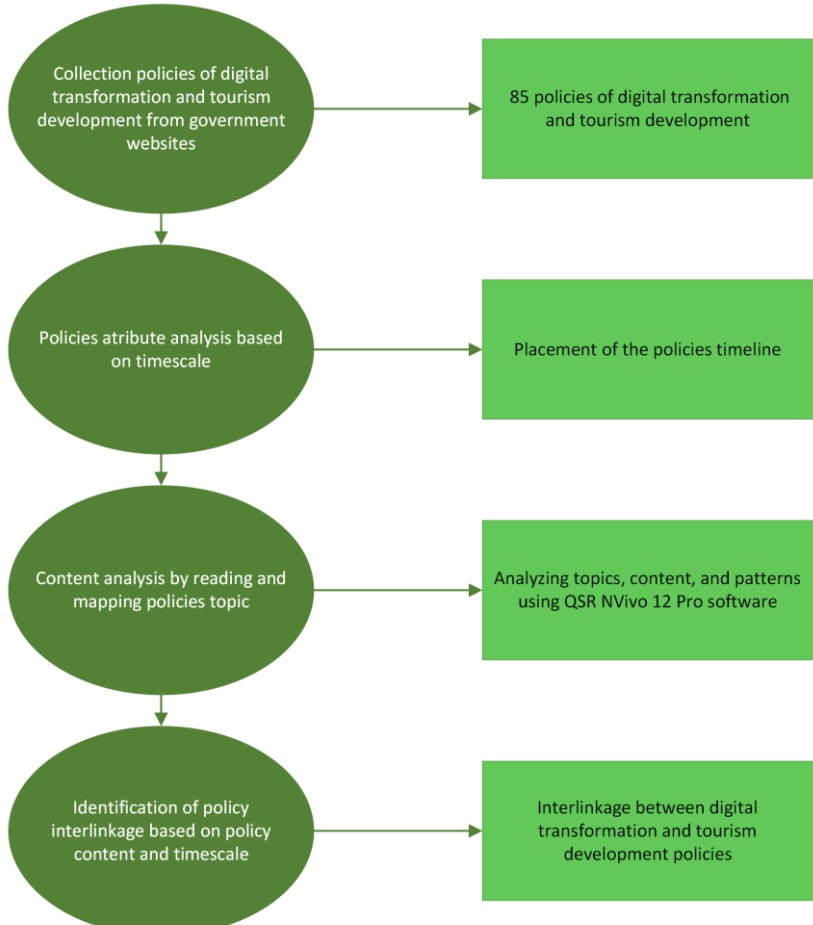

**Figure 2.** Steps for data collection and analysis.

First, the collection of digital transformation policies and tourism development in Indonesia is performed through government websites. Data collection was carried out on 17–19 May 2023. The policy is sourced from government websites, namely, the National Legal Documentation and Information Network (JDIH) in "jdihn.go.id (accessed on 17–19 May 2023)" and the JDIH Audit Board in "jdih.bpk.go.id (accessed on 17–19 May 2023)". Specifically for tourism policy, some data were withdrawn from the JDIH of the Ministry of Tourism and Creative Economy/Tourism and Creative Economy Agency (Kemenparekraf) in "jdih.kemenparekraf.go.id (accessed on 17–19 May 2023)". The websites was chosen to maintain validity based on policies taken from sources accessible to everyone [49]. Content analysis was used because it allows researchers to analyze directly, especially the data collected in qualitative (policy content) and quantitative (policy time) forms to improve research reliability [50]. The keywords used in collecting digital transformation policies included *"transformasi digital"*, *"elektronik"*, *"digital"*, *"teknologi"*, and *"elektronifikasi"* in Indonesian. Meanwhile, for the collection of tourism policies used two languages, namely, Indonesian, using the keywords "tourism development", "tourism", "tourism village", and, in Dutch, using the words *"reiziger"*, *"reiz"*, *"tourism"*, and *"tour"*.

Through this process, 95 Indonesian national policies with the theme of digital transformation and tourism development were found between 1879 and 2022. The following forms of policy are found, among others.

- Staatsblad (St)
- Law (UU)
- Government Regulation (PP)
- Presidential Regulation (Perpres)

- Ministerial Regulation (Permen): Tourism, Post, and Telecommunications (Parpostel); Culture and Tourism (Budpar); Tourism and Creative Economy (Parekraf); Tourism (Par); and Tourism and Creative Economy/Tourism and Creative Economy Agency of the Ministry (Parekraf/Baparekraf); Marine and Fisheries (KP); Environment and Forestry (LHK); Utilization of State Apparatus and Bureaucratic Reform (PanRB).
- Presidential Decree (Keppres)
- Ministerial Decree (Kepmen): Tourism and Creative Economy (Parekraf); Tourism (Par); and Tourism and Creative Economy/Tourism and Creative Economy Agency of the Ministry (Parekraf/Baparekraf); Utilization of State Apparatus and Bureaucratic Reform (PanRB).
- The Decision of State Institutions: Decision of the Regional Representative Council (KepDPD)
- Presidential Instruction (Inpres)

Based on the results of the policy reading, 87 policies related to digital transformation and tourism development were filtered, with 83 policies using Indonesian and 4 policies using Dutch. Second, timescale policy attribute analysis was conducted. Policies that have been read are then compiled based on historical policy time attributes. With this, the frequency of policies can also be known in a certain period. Third, content analysis by reading and mapping policy topics. A total of 87 filtered policies were then analyzed for topics, content, and policy patterns. The analysis was conducted separately between digital transformation policies and tourism development policies. Furthermore, in mapping and organizing the text of these policies, we used QSR Nvivo 12 software [47]. The coding process is carried out by specifying 'codes' and 'nodes' in the coding process, which then creates a relationship by reading text through a computer [51]. Nvivo has superior data management facilities because all data can be stored digitally and quickly recalled [52]

In order for the topics contained to be known, a last step is involved, where we identify policy linkage based on content and policy timescale. Based on the resulting timeline, analysis was then carried out to map the linkage between policies regarding digital transformation and tourism development. These relationships can be topics and content, as well as intersecting periods.

## 4. Results

### 4.1. Policy Timeline for Digital Transformation and Tourism Development in Indonesia

There are a total of 87 policies regarding digital transformation and tourism development in Indonesia (see Figure 3). The timeline shows that digital transformation policies will be set in Indonesia from 1997 to 2022. Indonesia has as many as 26 policies regarding digital transformation between 1997 and 2022. Meanwhile, policies regarding tourism development began to exist during the Dutch colonial era from 1879–2022, with 63 policies.

Based on Figure 3, during the COVID-19 pandemic in Indonesia, which lasted from 2020–2022, the trend of tourism development policies was higher than in previous years, namely, as many as 12 policies, while there were nine policies concerning digital transformation. Compare this to the 2017–2019 period, where tourism development policies had eight policies, and digital transformation had three.

Indonesia's digital transformation and tourism development policies were set in the same few years (see Table 1). These are 2000, 2003, 2008, 2009, 2012, 2014. 2016, 2018, 2019, 2020, 2021, and 2022.The number of policies issued in these years varies greatly, each in the one to six policies range. Likewise, the forms of policies given are: UU, Perpres, Keppres, Permen, Kepmen, and Decisions of State Institution.

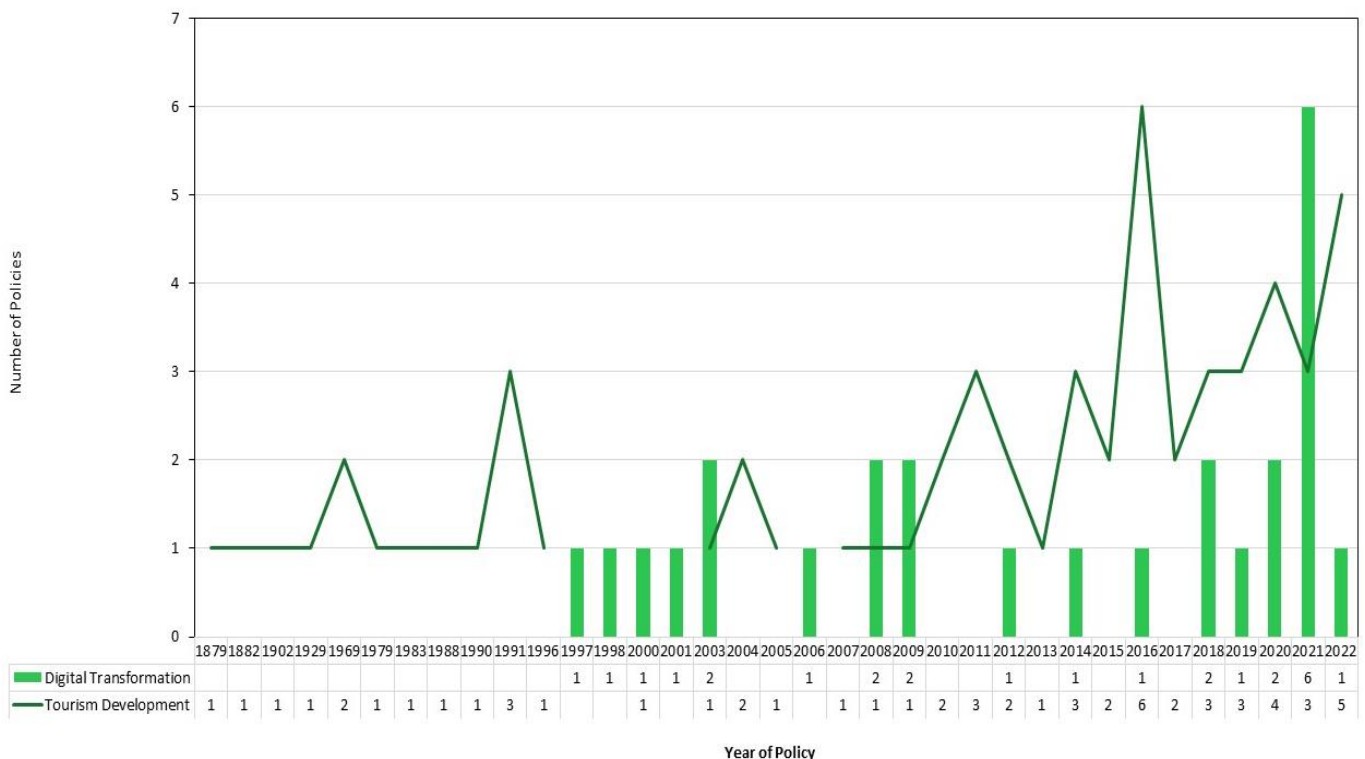

**Figure 3.** Policy Timeline for Digital Transformation and Tourism Development in Indonesia.

**Table 1.** Indonesia's digital transformation and tourism development policies were set in the same years.

| Digital Transformation | Year | Tourism Development |
| --- | --- | --- |
| Keppres 50 | 2000 | Keppres 11 |
| Inpres 3 and Keppres 9 | 2003 | Kepmenbudpar 1 |
| UU 11 and UU 14 | 2008 | Permenbudpar 4 |
| Keppres 5 and 38 | 2009 | UU 10 |
| PP 82 | 2012 | Permenparekraf 27 and 37 |
| Keppres 1 | 2014 | Perpres 63 and 64; and Permenpar 1 |
| UU 19 | 2016 | Perpres 50; PermenKP 14; Permenpar 9,10,14, 20; |
| Perpres 95 and KepmenpanRB 5 | 2018 | Kepmenpar 17; Perpres 14; and Permenpar 10 |
| Perpres 39 | 2019 | KepDPD 95; PermenLHK 8; and Permenpar 10 |
| PermenpanRB 5 and 59 | 2020 | PermenKP 93; PermenLHK 13; Permenparekraf/Baparekraf 8 and 13 |
| KepmenpanRB 962; KepmenpanRB 963; KepmenpanRB 965; KepmenpanRB 1503; Keppres 3; and Permendagri 56 | 2021 | Permenparekraf/Baparekraf 3, 5, and 9 |
| Perpres 1 | 2022 | Permenparekraf/Baparekraf 2, 4, 12, 14; and Perpres 26 |

The 87 policies obtained are St, UU, PP, Perpres, Keppres, Permen, Kepmen, and Decree of State Institutions (see Figure 4). The President issued 28 policies through regulations, decrees, and instructions from 1969–2022. The most often seen policy form is Permen, with 33 policies. These policies are issued by the minister in charge of tourism, state apparatus and bureaucratic reform, marine and fisheries, environment and forestry, and internal affairs. Then followed by the Keppres with as many as 14 policies; Kepmen with 11 policies. Perpres had as many as nine policies; UU, PP, and Inpres each had five policies; St had as many as four policies; and Decrees of State Institutions had as many as two policies.

The Minister of Tourism and Creative Economy/Head of the Tourism and Creative Economy Agency (Menparekraf/Kabaparekraf), who leads tourism and creative economy affairs in Indonesia, had, for some time, experienced changes in nomenclature and changes in matters in the fields as stated in Figure 5. Based on the findings of 30 policies on tourism development issued by the Minister in charge of tourism, there have been five changes in ministerial nomenclature. In two policies in Decree No. 48 and No. 82 of 1991, this Minister

was named the Minister of Tourism, Postal Service, and Telecommunications, who led the Ministry of Tourism, Post, and Telecommunications. Based on Presidential Decree 25/1990, the Minister has the main task of carrying out part of the general duties of government and development in the fields of tourism, post, and telecommunications responsible to the President.

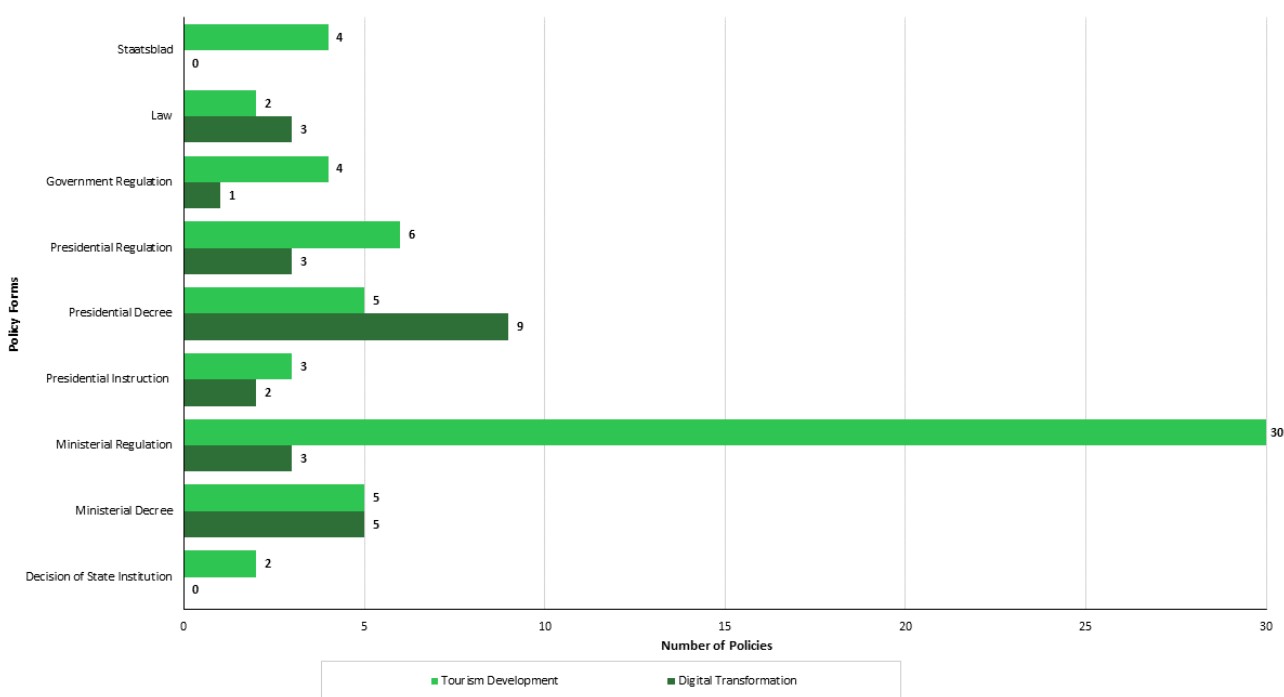

**Figure 4.** Digital Transformation and Tourism Development Policy Forms.

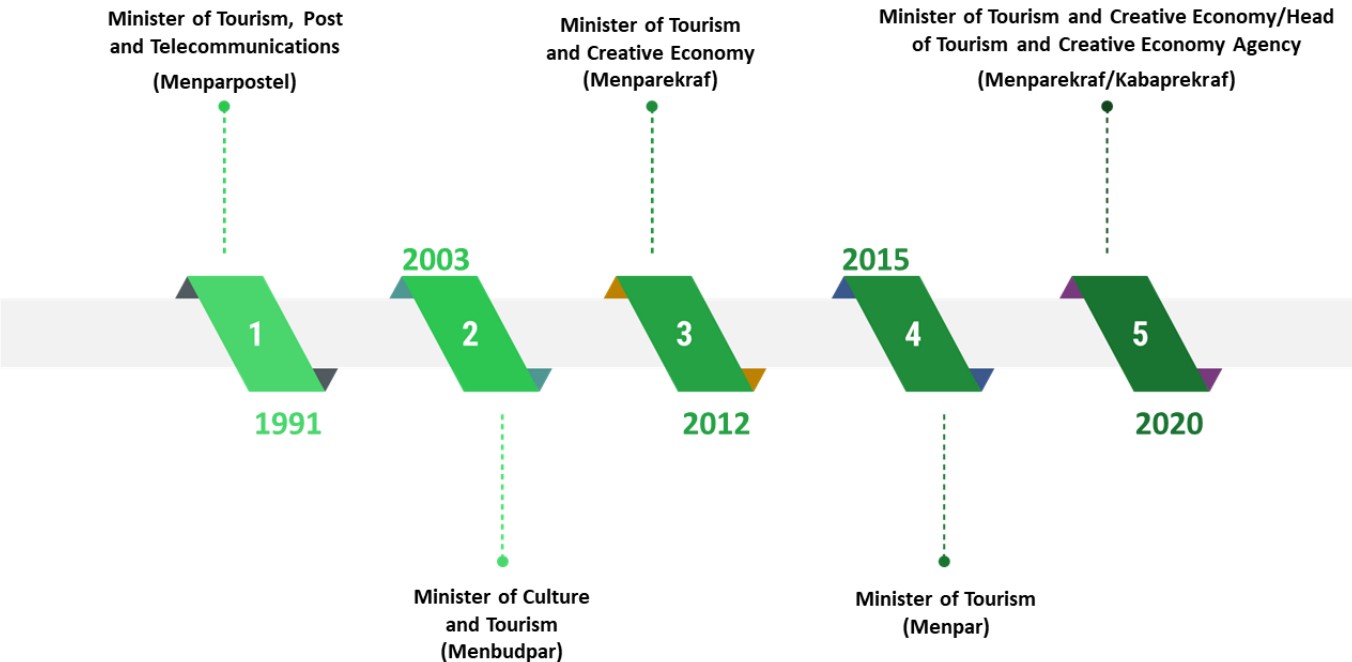

**Figure 5.** Changes in the nomenclature of the Ministry of Tourism and Creative Economy/Tourism and the Creative Economy Agency, which are based on development tourism policies.

Then, the second phase changed to the Minister of Culture and Tourism listed in five Permen and four Kepmen from 2003–2011. The nomenclature was first stated in

Kepemenbudpar 1/2003. Based on Presidential Decree 103/2001, the Minister leads the Culture and Tourism Development Agency to assist the President in formulating policies and coordination in the field of culture and tourism.

Furthermore, the Permen in 2012 changed it to Minister of Tourism and Creative Economy, first listed in Permenparekraf 27/2012. Based on Presidential Regulation 92/2011, the Minister leads the Ministry of Tourism and Creative Economy and assists the President in both fields. A fourth nomenclature change to Minister of Tourism, as stated in eight Ministerial Regulations and one Ministerial Decree in the 2015–2019 period, was conducted. This change first appeared in the policy found, namely, Permenpar 22/2015. They assist the President in planning tourism affairs through the Ministry of Tourism by Presidential Regulation 19/2015.

It was last changed in 2020–2022, listed in eight regulations, to become the Minister of Tourism and Creative Economy/Tourism and Creative Economy Agency. The first time the change appeared was in a policy, namely, Permenparekraf/Baparekraf 8/2020. This change in nomenclature is based on Presidential Decree No. 96 and No. 97 of 2019, promulgated on 31 December 2019. There is an addition to the affairs, namely, in the area of the creative economy. The scope of the creative economy includes application, interior design, visual communication design, fashion, film, photography, advertising, television and radio, game development, and others.

Based on this explanation, there have been five changes in the nomenclature of ministers in charge of tourism in Indonesia, either singly or in combination with other fields. Only once was a single minister in charge of tourism under the name of the Minister of Tourism. In 2001, the tourism sector was managed by non-departmental government agencies. Additionally, the same is the case with the current Baparekraf, which was present in 2020 by carrying out operational tasks to coordinate and synchronize tourism and creative economy policies that the Minister of Tourism and Creative Economy does not hold.

### 4.2. Policy Content Analysis

#### 4.2.1. Tourism Development

In the 59 policies found in the range of 1879–2022, tourism development policies have various discussions. The first topic is *accessibility*, which is supported by 13 policies or 22.03% of the total tourism development policies found. *Accessibility* is related to facilitating tourist movements on their way through land, sea, and air transportation modes [27]. This can be seen in Table 2 through a tabulation process based on policy content analysis. This node is associated with St 54/1882, St 441/1901, St 70/1929, and St 86/1879 during the Dutch colonial period in Indonesia. These four policies discuss transportation modes and costs for travelers. While, in St 54/1882, the size and weight of the luggage of tourists were regulated, namely, cargo, including luggage up to 10 kg with a length of 0.5 m, a width of 0.3 m, and a height of 0.2 m. Then, there is Presidential Decree 39/1988, which regulates flight entrances and seaport entrances, as well as visa rules for foreign tourists.

*Accessibility* is also listed as one of the criteria in determining leading tourist destinations in Permenbudpar 37/2007. Then, Permenpar 1/2014 and Permenpar 10/2016 regulate regional tourism development concerning destination *accessibility*. Meanwhile, in PermenLHK 76/2015 and KepDPD 16/2019, the arrangement and management of tourism zones are examined. The last policies are Permenparekraf 13/2020 and Permenparekraf 4/2022, issued during the COVID-19 period regarding *accessibility* to and in tourist destinations that are in accordance with hygiene standards, health, safety, and environmental sustainability and guidelines for the use of non-physical special allocation funds for tourism services in 2022. Therefore, the 13 policies have a relationship with the topic of *accessibility*.

**Table 2.** Content analysis of tourism development policies.

| Topic | Number of Policies | Policies |
|---|---|---|
| *Accessibility* | 13 | St 54/1882; St 441/1901; St 70/1929; St 86/1879; Keppres 15/1983; Keppres 39/1988; Permenbudpar 37/2007; Permenpar 1/2014; PermenLHK 76/2015; Permenpar 10/2016; KepDPD 16/2019; Permenparekraf 4/2022; Permenparekraf 13/2020 |
| *Activity* | 5 | Inpres 3/1991; Kepmenparpostel 46/1991; Inpres 1/2005; Permenbudpar 37/2007; and Permenparekraf 13/2020 |
| *Amenity* | 17 | PP 4/1979; Keppres 15/1983; UU 9/1990; Kepmenparpostel 46/1991; Inpres 16/2005; Permenbudpar 37/2007; UU 10/2009; PP 20/2011; Permenpar 1/2014; PermenLHK 76/2015; PermenKP 14/2016; Permenpar 10/2016; Kep DPD 16/2019; PermenKP 93/2020; PermenLHK 13/2020; Permenparekraf/Baparekraf 4/2022, 12/2022, dan 13/2020 |
| *Attraction* | 13 | Permenbudpar 67/2004; Inpres 1/2005; PP 20/2011; Perpres 63/2014; PermenLHK 76/2015; PermenKP 14/2016; PermenLHK 8/2019; Permenpar 10/2019; PermenKP 93/2020; PermenLHK 13/2020; Permenpar 14/2016; and Permenparekraf/Baparekraf 9/2021 |
| *Environmental Aspect* | 11 | Permenbudpar 67/2004; Perpres 63/2014; PermenLHK 76/2015; PermenKP 14/2016; Permenpar 9/2016 and 14/2016; PermenLHK 8/2019; Permenpar 10/2019; PermenKP 93/2020; PermenLHK 13/2020; Permenparekraf/Baparekraf 9/2021 |
| *Stakeholder's Role* | 42 | Inpres 9/1969; PP 4/1979; UU 9/1990; Inpres 3/1991; Kepmenparpostel 46/1991 and 82/1991; Kepmenbudpar 67/2004; Inpres 1/2005; Permenbudpar 37/2007 and 26/2010; UU 10/2009; Permenbudpar 26/2010; PP 36/2010; Keppres 22/2011; Permenbudpar 18/2011; Permenparekraf 27/2012 and 37/2017; KepDPD 47/2013; Permenpar 1/2014; Perpres 53/2014 and 64/2014; Permenpar 10/2016, 14/2016, 20/2016, 22/2016; Perpres 50/2016; Permenpar 13/2017; Perpres 14/2018; KEpDPD 16/2019; PermenLHK 10/2019; Permenpar 10/2019; PermenKP 93/2020; PermenLHK 13/2020; Permenparekraf/Baparekraf 8/2020, 13/2020, 3/2021, 6/2021, 9/2021, 2/2022, 4/2022, 12/2022, and 14/2022 |
| *Task Force* | 8 | Keppres 30/1969; Kepmenbudpar 1/2003; Keppres 22/2011; Permenparekraf 37/2012; Perpres 64/2014 and 40/2017; Kepmenpar 17/2018; and Perpres 14/2018 |
| *Technology Adoption* | 23 | Kepmenparpostel 82/1991; PP 67/1996, 36/2010, and 20/2011; Permenparekraf 27/2012; KepDPD 47/2013; Perpres 63/2014; PermenKP 14/2016; Permenpar 1/2014, 22/2015, 20/2016, 13/2017, 10/2018; KepDPD 16/2019; PermenLHK 8/2019; PermenKP 93/2020; PermenLHK 13/2020; Permenparekraf/Baparekraf 8/2020, 13/2020, 3/2021, 9/2021, 2/2022, and 4/2022 |
| *Tourism Business* | 7 | PP 67/1996; UU 10/2009; PP 36/2010; Permenpar 13/2017 and 10/2018; PermenLHK 8/2019; and Permenparekraf/Baparekraf 13/2020 |
| *Tourism Standardization* | 17 | Permenbudpar 37/2007; UU 10/2009; PermenKP 14/2015; Permenpar 9/2016 and 13/2017; Permen KP 93/2020; Permenpar 14/2016; PermenLHK 13/2020; Permenparekraf/Baparekraf 8/2020, 13/2020, 3/2021, 6/2021, 9/2021, 2/2022, 4/2022, and 12/2022 |

Second, discussions on *activity* and efforts to support tourism events that have been carried out in Indonesia are also regulated by policy. Policies concerning the *activity* topic are found in five policies: Presidential Instruction 3/1991, Kepmenparpostel 46/1991, Presidential Instruction 1/2005, Permenbudpar 37/2007, and Permenparekraf 13/2020. The form of activity is written as in Kepmenparpostel 82/1991, regarding support for the 40th PATA (Pacific Asia Travel Association) Conference in 1991 in Bali with the theme "Enrich The Environment". In addition, 16/2005 listed efforts to develop Indonesian tourism, with the theme "Indonesia Ultimate in Diversity" for abroad and "Know Your Country, Love Your Country, Let's Sightseeing Explore the Archipelago" for the domestic audience. Meanwhile, Presidential Instruction 3/1991 contains a national Tourism Awareness Campaign to mobilize national potential as a sustainable effort to support the implementation of the ASEAN (Association of South-East Asian Nations) Tourism Visit Year 1992. Meanwhile, in Permenbudpar 37/2007, issues concerning activities carried out by regions to become leading tourism destinations whose provisions are regulated through this policy are examined. Then, in Permenparekraf 13/2020, which was promulgated during the COVID-19 pandemic, it stipulates activities related to tourism, cleanliness, health, safety, and environmental sustainability of the sector during the COVID-19 handling period. Therefore, these policies are associated with the topic of *activity*.

The third topic is *amenity*, associated with 17 policies or 28.81% of the total tourism development policies found. *Amenity* is a component that combines the needs of tourists

with their facilities and infrastructure, such as tourist information centers, hospitals, and food courts [27]. PP 4/1979 states that guidance in the field of tourism is handed over to the provincial government with the task and authority to regulate the affairs of tourist *attraction*s, accommodation, tourism businesses, and promotion of local tourism. In addition, there is a tourism policy regarding facilities in tourist destinations regulated in Law 9/1990, later amended into Law 10/2009. The following policy is; Kepmenparpostel 46/1991, which regulates the development of tourism facilities, and infrastructure is carried out based on the General Plan of Spatial Planning (RUTR), Environmental Impact Assessment (AMDAL), and the preparation of the presentation of environmental information, and each tourism area must have cultural arts performance and infrastructure development using local labor sources.

The next *amenity* policy involves Presidential Instruction 1/2005, Permenbudpar 37/2007, and PP 20/2011, which discuss developing tourism facilities and public facilities in tourist destinations nationally. Then, the policies of Permenpar 1/2014 and 10/2016 stipulate the implementation of deconcentration activities and assistance tasks of the Ministry of Tourism regarding the mechanism for building facilities in tourist destinations related to reporting and accountability, handover of goods, inspection, guidance, and supervision. The following policies, PermenLHK policy 76/2015, PermenKP 14/2016, Kep DPD 16/2019, PermenKP 93/2020, and PermenLHK 13/2020 regulate approaches and activities, criteria, and procedures for determination, action plans, implementation, monitoring, and evaluation of zone utilization for tourism and marine villages. The latest policy of Permenparekraf/Baparekraf 13/2020, 4/2022, and 12/2022 stipulated, during the COVID-19 pandemic, regulates the standardization and facilitation of financing for Indonesian National Standards regarding cleanliness, health, safety, and environmental sustainability, as well as technical guidelines for the utilization of special allocation funds for facilities and infrastructure in tourist destinations. Therefore, the 17 policies are associated with the topic of *amenity*.

On the other hand, the topic of *attraction*, namely, the attractiveness of a tourist *attraction*, was contained in 13 policies or 22.03% of the total tourism development policies. *Attraction* can be interpreted as an object that becomes a tourist *attraction* that can attract interest and influence travelers' travel decisions [27]. In Permenbudpar 67/2004, the policy specifically provides guidelines for tourism development in small islands based on the principle of balance, the principle of community participation, the principle of conservation, the principle of integration, and the principle of law enforcement. Meanwhile, Presidential Instruction 16/2005 stipulates cultural and tourism development policies with the optimization of national tourism and the utilization of resources for tourism development. Then, PP 20/2011 and Presidential Regulation, 63/2014 explains the development of national tourism by covering tourism destinations in order to realize Indonesia as a world-class tourism destination country and regulating tourism supervision and control related to various negative impacts in the broader community.

Other policies, supporting the *attraction* topic, are PermenLHK 76/2015, 8/2019, and 13/2020, regulating the development of natural tourism *attraction*s in wildlife reserves, national parks, botanical forest parks, and nature tourism parks, as well as the development of natural forest tourism. Then, in the marine and fisheries sector, there are PermenKP 14/2016 and PermenKP 93/2020 policies, which regulate the determination of marine protected areas for aquatic nature tourism and *attraction* management in marine tourism villages. Finally, Permenpar policy 14/2016, Permenpar 10/2019, and Permenparekraf/Baparekraf 9/2021 regulate the management of *attraction*s in sustainable tourism destinations (socioeconomically, culturally, and environmentally) and the management of tourism *attraction* crises. Therefore, the 13 policies support the topic of *attraction*.

The *tourism business* is also a topic that is widely found in tourism development policies contained in seven policies. This is also the case in the PP 67/1996 policy, which concerns tourism development. Meanwhile, Law 10/2009 regulates, in general, the terms of the registration and the policy of tourism businesses. Then, PP 36/2010 and PermenLHK

8/2019 concern permits, obligations, and rights, as well as cooperation in the exploitation of natural tourism in wildlife reserves, national parks, botanical forest parks, and nature tourism parks. Furthermore, Permenpar 13/2017 regulates management support for deconcentration management, industrial development, and tourism institutions. Then, a more specific policy discusses electronically integrated tourism business licensing regulated by Permenpar 10/2018. Finally, the Permenparekraf/Baparekraf 13/2020 policy holds standards and certifications related to health and the environment during the COVID-19 pandemic for *tourism businesses*. Therefore, the seven policies are associated with the topic of *tourism business.*

On the other hand, *environmental aspects* related to tourism businesses and activities are also not spared from policies, namely, 11 policies, or 18.64% of the total tourism development policies discussing this issue. Presidential Regulation 63/2014 states that the government and local governments are obliged to carry out supervision to prevent adverse impacts caused by tourism. Then, in PermenLHK 76/2015, 8/2019, and 13/2020, there are regulations for the structuring, monitoring, and evaluation of management blocks, national park management criteria related to the provisions for its business, and requirements for the construction of tourist facilities and infrastructure.

Then, PermenKP 14/2016 and 93/2020 regulate the category of marine protected areas for aquatic nature tourism and the determination of maritime protected areas for preserving maritime customs and culture. Furthermore, Permenpar 9/2016 and 10/2019 policies regulate the technical procedures for handling tourism crises, preparedness and mitigation, emergency response, recovery, and normalization.

Then, three policies that specifically regulate sustainability and sustainable tourism. in Indonesia are proposed. In this discussion, there is a term, sustainable tourism, which is based on those regulations listed in Permenbudpar 67/2004, Permenpar 14/2016, and Permenparekraf/Baparekraf 9/2021. Permenbudpar 67/2004 regulates the utilization of the potential of tourism resources of small islands through the management of sustainable tourism activities to improve community welfare in the economic and cultural fields, as well as regional development. In this policy, sustainable tourism is stated as the implementation of responsible tourism that meets the needs and aspirations of humans today, without sacrificing the potential to meet human needs and aspirations in the future, by applying the principles, economically feasible, environmentally viable, socially acceptable, socially acceptable, and technologically appropriate.

Sustainable tourism in Permenpar 14/2016 contains guidelines for sustainable tourism destination management, economic utilization for local communities, cultural preservation for the community and visitors, and environmental preservation. The guidelines are aligned with indicators from the United Nations World Tourism Organization (UNWTO) and have been recognized by the Global Sustainable Tourism Council (GSTC). Then, the policy was updated in Permenparekraf/Baparekraf 9/2021, which regulates sustainable management, social and economic sustainability, cultural sustainability, and environmental sustainability. In this new regulation, more emphasis is placed on the issues of cleanliness, health, safety, and environmental sustainability, which are anticipatory measures against the spread of COVID-19.

Tourism development also tries to elaborate on aspects of technology grouped in the topic of *technology adoption,* found in 23 policies or 38.98% of the total tourism development policies. Starting with the policy of Kepmenparpostel 82/1991, which regulates information systems development in tourism, post, and telecommunications management. Then, PP 67/1996, 36/2010, and 20/2011 concern the use of electronic media and networks, information, and technology included in tourism management. Furthermore, Permenparekraf 27/2012 regulates the installation of advertisements on internet media, internet search engines, or regional tourism websites.

Policies on other *technology adoption* topic are KepDPD 47/2013 and 16/2019 regulating internet sites and social media use for tourism promotion and partnerships with Google applications, Baidu, Alibaba, Expedient, Grab, Traveloka, and Tiket.com (accessed

on 21 May 2023). Then, Presidential Regulation 63/2014 regulates the prevention and mitigation of tourism activities that cause negative impacts carried out by one of them by using methods that follow the development of science and technology. Furthermore, the policies of Permenpar 1/2014, 22/2015, 20/2016, 13/2017, and 10/2018 regulate the Tourism Information Center (TIC), as well as the use of technology, internet, social media, and licensing using Online Single Submission (OSS) for tourism business actors.

In addition, there are policies in PermenLHK 8/2019 and 13/2020 regarding efforts to provide information services for natural tourism services in the form of delivering data, news, features, photos, videos, and research results on tourism that are disseminated in the form of printed or electronic materials, as well as TIC in forest tourism destinations. Then, the policies of PermenKP 14/2016 and 93/2020 regulate communication networks in tourist areas and action plans for marine tourism villages related to access to technology and information and promotion through electronic media. Furthermore, Permenparekraf/Baparekraf 8/2020, 13/2020, 3/2021, 9/2021, 2/2022, and 4/2022 issued during the COVID-19 pandemic provide more space in the use of technology by facilitating digitalization training, development of the digital economy and creative products, development of Tourism Information System (TIS) in the form of online and offline applications or software, internet access facilities, and promotion via electronic media. Therefore, based on the previous explanation, 23 are associated with technology adoption.

The next topic is the *task force* supported by eight policies, or 15.25% of the total tourism development policies found. Here are the task forces regarding tourism development that have been formed in Indonesia to do certain jobs.

1.　National Tourism Advisory Council

Based on Presidential Decree 30/1969, in article 2, it explains that the National Tourism Advisory Council consists of:

- The Minister of State for Economy, Finance and Industry, as Chairman, who is concurrently a member
- The Minister of Transportation, a member
- The Minister of Trade, a member
- The Minister of Finance, a member
- The Minister of Industry, a member
- The Minister of Education and Culture, a member
- The Minister of Public Works and Electricity, a member
- The The Minister of Justice, a member
- The Minister of Defense and Security, a member
- The Minister of Foreign Affairs, a member
- The Minister of Home Affairs, a member
- The Minister of Social Affairs, a member
- The Minister of Information, a member
- The Central Bank Governors, members
- The Chairman of the National Development Planning Agency, a member

2.　Working Group to Support the Implementation of a National Tourism Recovery Working Group

Based on Kepmenbudpar 1/2003, this group promotes tourism and tourism events. The Minister of Culture and Tourism serves as the director of this task force.

3.　Indonesian Tourism Promotion Agency

Based on Presidential Decree 22/2011 and Permenparekraf 37/2012, the Indonesian Tourism Promotion Agency is a private and independent institution domiciled in the national capital. The Tourism Promotion Agency has the task of improving the image of Indonesian tourism, increasing foreign tourist arrivals and foreign exchange receipts, increasing domestic tourist arrivals and spending, raising funding from sources other than the State Budget and Regional Budget following the provisions of laws and regulations,

and conducting research in the context of business development and tourism business. In carrying out its duties, this task force has the function of a tourism promotion coordinator and partner of the central and local governments.

4. Cross-Sector Strategic Coordination Team for Tourism Implementation

Based on Presidential Decrees 64/2014, 40/2017, and 14/2018, the Tourism Coordination Team is underneath the authority of and responsible to the President. The Vice President leads the *task force,* and the daily chairman is headed by the Minister of Tourism. In Kepmenpar 17/2018, this task force is assisted by a daily implementation team that is responsible and reports the results of its activities to the Vice President.

The next topic concerns the *stakeholders' role* being contained in 43 policies or 72.88% of all tourism development policies. The *stakeholder's role* in this policy refers to the *central government, local governments, the private sector, and the community*, according to Figure 6 using Nvivo 12 Pro analysis.

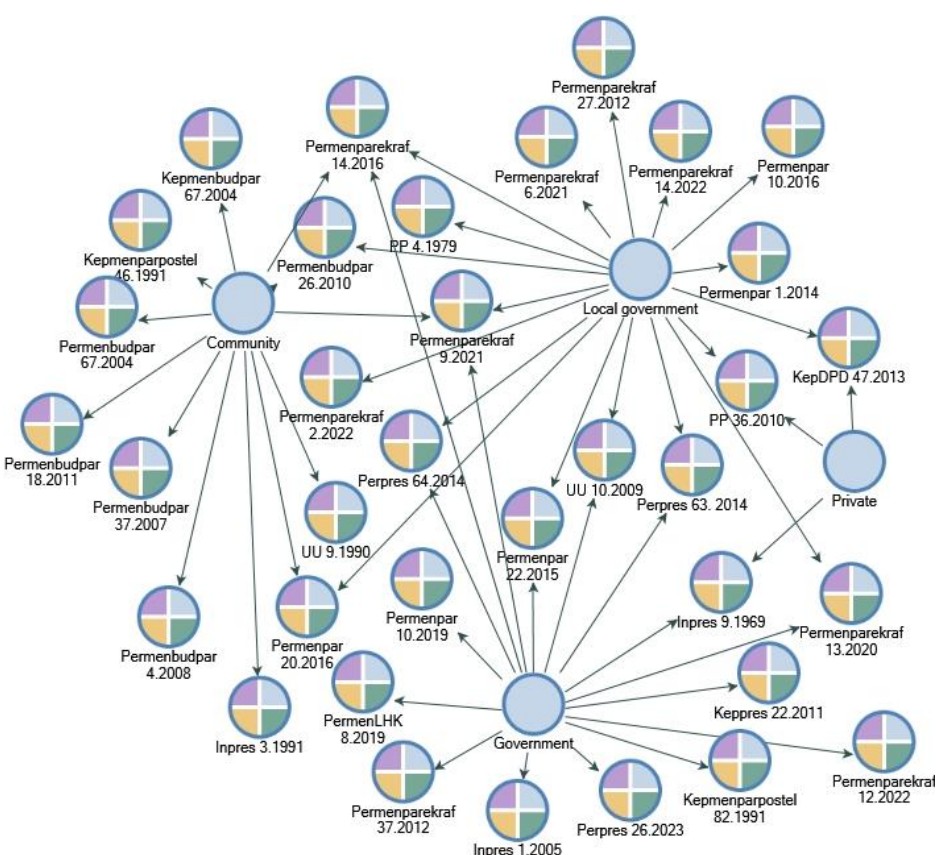

**Figure 6.** The stakeholders in tourism development policies.

The *government* plays a role in distributing benefits and managing tourism fairly for each stakeholder, which is implemented to suppress externalities for tourism practices and regulate their use for the economy and society [27,53]. The role of central government is listed in 15 tourism development policies. The policy is similar to Law 10/2009, which regulates general related to tourism in Indonesia. This rule is the master rule for tourism in Indonesia, which has various derivative rules, such as PP, Permen, Keppres, Inpres, and so on. In addition, the role of *local government* is contained in 18 policies on the topic of stakeholder's role. In three ministerial regulations in charge of tourism, namely, Permenparekraf 27/2012, Permenpar 1/2014, Permenparekraf/Baparekraf 9/2021, and 14/2022, which regulate the implementation of deconcentration activities and assistance tasks in the regions.

In addition, the role of the *community* is listed in 11 policies of the tourism development policy. One of these is Permenbudpar 4/2008, which regulates tourism awareness, a condition that describes the participation and support of all components of society in encouraging the realization of a conducive climate for the growth and development of tourism in a destination or region. Concerning this policy, it confirms the role of the community in tourism development. Then, the role of the *private sector* is listed in three policies, namely, Presidential Instruction 9/1969, PP 36/2010, and KepDPD 47/2013. The regulation is about tourism development efforts, private nature tourism business, and the involvement of private parties, such as multinational start-up companies as government partners.

The last topic is *tourism standardization*, supported by 17 policies or as much as 28.81%. Permenbudpar 37/2007 PermenKP 14/2015, PermenLHK 13/2020, Permenparekraf/Baparekraf 8/2020, 13/2020, 3/2021, 4/2022, and 12/2022 regulate criteria, standards, and certification in tourism development for the use of funds and classification of tourist destinations. Then, Law 10/2009, Permenpar 9/2016 and 13/2017, PermenKP 93/2020, and Permenparekraf/Baparekraf, 6/2021, 2/2022, and 12/2022 regulate guidelines and standardization in the implementation of deconcentration and assistance tasks regarding tourism in the regions. The following policy is Permenpar 14/2016 and Permenparekraf/Baparekraf 9/2021, which regulate criteria and guidelines in the implementation of sustainable tourism.

These regulations are based on the explanation of 59 policies regarding tourism development in Indonesia from 1879–2022 into nine topics of analysis. These topics are *accessibility, activity, amenity, attraction, environmental aspect, stakeholder role, task force, technology adoption, tourism business,* and *tourism standardization*. From the analysis, it was found that the topics with the largest number of policy associations were contained in the stakeholders' role, while the topic with the smallest number of policy associations was activity in tourism in Indonesia.

4.2.2. Digital Transformation

Various topics were found in 26 policies regarding digital transformation in Indonesia between 1997–2022. This can be seen in Figure 7 through topic mapping processed using NVivo 12. First, the topic of *community digital literacy* is contained in two policies. These policies are UU 11/2008 and Inpres 2001. Based on this policy, digital transformation is carried out to educate people as part of the world information society and to open the broadest opportunity for everyone to advance their thinking and abilities in the use and utilization of information technology. In addition, it is also an effort to expand community involvement in the development of information and technology.

The second topic regarding the role of *government* was found in 19 policies or 73.08% of the total policies regarding digital transformation found. Presidential Regulation 95/2018 regulates the governance and management of the Electronic-Based Government System (SPBE) in Indonesia, with the establishment of a master plan coordinated by the minister who organizes government affairs in the field of national development planning. In addition, a policy of Perppres 39/2019 regulates the central government's role in managing government data to produce accurate, up-to-date, integrated, and accountable data, as well as easily accessible and shared with the participation of state institutions and other public legal entities. In addition, the government is committed to utilizing technology to prevent the digital divide with other countries.

Third, the topic of *local government* contained 11 policies, or 42.31% of the total policies, which played a role in implementing digital transformation policies from the national level to be implemented at the regional level. As in the policy of Presidential Decree 3/2021, which regulates the formation *of a task force* formed to encourage the implementation of the electronification of local government transactions and support the development of digital payment transactions for the community. In addition, *local government* in the Ministry of Home Affairs 56/2021 regulates the expansion of digitalization and electronification of local government transactions at the provincial and district/city levels.

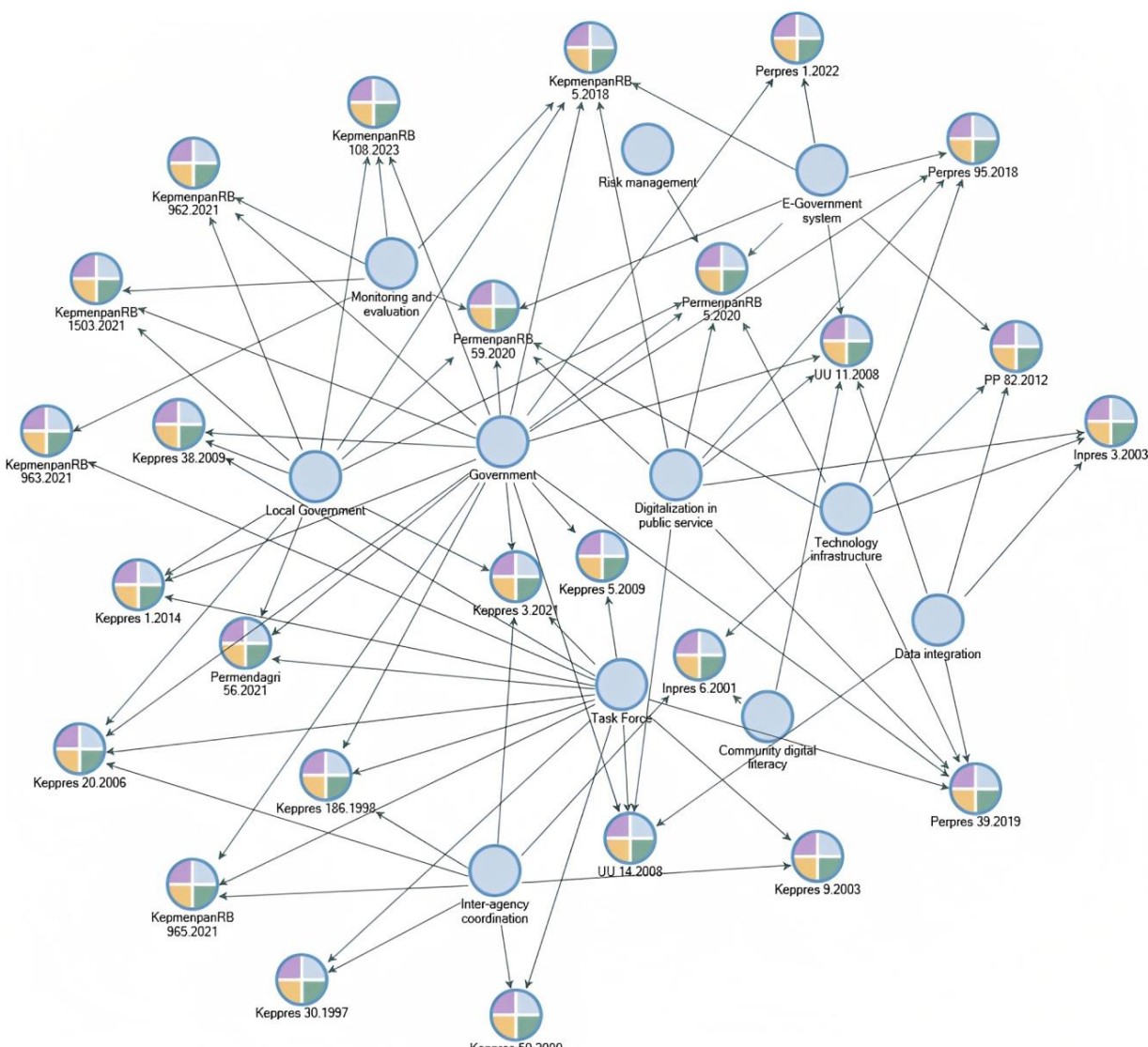

**Figure 7.** Content analysis of digital transformation policies.

Digital transformation in the public perspective also concerns the term *e-government*, which is carried out to improve service quality, increase accessibility to services and information, and increase the effectiveness of transactions that benefit public organizations and service users [54]. In *e-government*, mapping using Nvivo 12 Pro is divided into five subtopics: *data integration, digitalization in public services, e-government system, monitoring and evaluation, risk management*, and *technology infrastructure*.

The subtopic *of data integration* is found in five policies that address it. One of them is Presidential Regulation 39/2019, which specifically discusses Satu Data Indonesia (SDI/One Indonesian Data) to produce accurate, up to date, integrated, accountable, and easily accessible and shared data between agencies and governments. Then, subtopics on *digitalization in public services* are listed in eight policies. Digital transformation in public service is integrating technology into activities to support the sustainability of public organizations more efficiently [55].

Public services in the policy include government transactions, access and use of public information, and other electronic services. Meanwhile, the subtopic on the *e-government system* is contained in seven central government policies. *The E-Government system* in Indonesia is called the Electronic-Based Government System (SPBE), under the policy of Presidential Regulation 95/2018 as the basis for implementing e-government [56]. In the

implementation of *e-government,* there is one supporting stage, namely, *monitoring and evaluation,* which is contained in six policies. Monitoring and evaluation are carried out based on established guidelines and instruments. Furthermore, producing evaluation results for central and regional government agencies on implementing SPBE.

In addition, there is a topic on risk management in the implementation of *e-government,* which is contained in one policy, namely, PermenpanRB 5/2020, concerning risk management guidelines. The guidelines are used to guide central and regional agencies in implementing SPBE based on their respective characteristics. SPBE risk is an opportunity that can affect the achievement of SPBE objectives. The existence of infrastructure also supports the success of e-government. The subtopic *of technology infrastructure was* contained in seven policies. The infrastructure in question is the development, availability, and *accessibility* of facilities and infrastructure supporting *e-government.*

Finally, an inter-agency coordination subtopic is contained in eight policies. Government agencies carry out inter-agency coordination horizontally and vertically with local governments. One form of this coordination is the establishment of *task forces* across agencies and government levels to support digital transformation. There are various *task forces* found in digital transformation policies.

1.    Indonesian Telematics Coordination Team

This task force is found in the oldest policy found in this study, namely, Presidential Decree 30/1997. Then, it was updated sequentially by Presidential Decrees 186/1998, 50/2020, and 9/2003. The Indonesian Telematics Coordination Team is carried out by a secretariat functionally under the State Minister of Communication and Information. Then, to support the smooth implementation of tasks, a Working Group was formed from related agencies, experts, observers, the business world, professional institutions, universities, the telematics community, and the community.

2.    National Information and Communication Technology (ICT) Council

Based on Presidential Decrees 20/2006, 5/2009, 38/2009, and 1/2014, the National ICT Council has the task of formulating general policies and strategic directions for national development, conducting studies in solving strategic problems, national coordination, and approving the implementation of technology and communication programs. The President chairs the ICT Council and has a daily chairman of the Minister of National Development Planning/Head of the National Development Planning Agency.

3.    One Indonesian Data Organizer

This is based on Presidential Regulation 39/2019, which regulates the One Indonesian Data program and establishes a *task force* for One Indonesian Data Organizers. At the central level, the *task force* consists of a steering board, a central-level data trustee, a central-level trustee, and a central-level data producer.

4.    SPBE Monitoring and Evaluation Implementation Team

This *task force* was formed based on KepmenpanRB 963/2021, which has the task of setting policy directions, monitoring implementation, and providing direction to the implementation team in the implementation of SPBE monitoring. This *task force* comprises a steering team, an implementation team, and an external assessor team.

5.    SPBE Coordination Team

Based on KepmenpanRB 965/2021, a *task force* was formed for the SPBE Coordination Team to formulate and synchronize policies, coordinate, and synchronize policy implementation, and monitor and evaluate SPBE implementation. The SPBE Coordination Team consists of the National SPBE Coordination Team, the National SPBE Coordination Team at the Intermediate Level, and the National SPBE Coordination Team at the Primary Level.

6.    Task Force for the Acceleration and Expansion of Regional Digitalization (Satgas P2DD)

The establishment of this *task force* is based on the policy of Presidential Decree 3/2021, chaired by the Coordinating Minister for Economic Affairs with the following members.

- Governor of Bank Indonesia
- Minister of Home Affairs
- Minister of Finance
- Minister of Communication and Information Technology
- Minister of State Secretary
- Minister of State Apparatus Empowerment and Bureaucratic Reform
- Minister of National Development Planning/Head of Agency

In addition, provincial and district/city governments also formed P2DD Teams chaired by regional heads.

7.   Regional Digitalization Acceleration and Expansion Team (TP2DD)

Based on Permendagri 56/2021, TP2DD was formed, a coordination forum between agencies and related stakeholders at the provincial and district/city levels to encourage innovation and accelerate and expand the implementation of electronification. The structure and membership of the provincial TP2D are chaired by the governor and the provincial regional secretary as the daily chief executive.

Based on the previous explanation, 26 digital transformation policies in Indonesia from 1997–2022 are grouped into *community digital literacy, government, local government, e-government,* and *task force.* For the topic of *e-government,* we divided it into several subtopics, namely, *data integration, digitalization in public services, e-government system, monitoring and evaluation, risk management,* and *technology infrastructure.* This analysis found that the topic associated with policy the most was the topic *of government.* While the topic with the least policy support is the *risk management* subtopic, part of the *e-government* topic.

### 4.3. Linkage Topic

#### 4.3.1. Digital Transformation and Tourism Policies Based on the President's Term of Office

Through the policies collected, 28 out of 87 or 32.18% of the total policies in the 55 years since 1969, Indonesia has been led by five different Presidents. Here are the Presidents and policies on digital transformation and tourism development issued by presidential terms (see Figure 8).

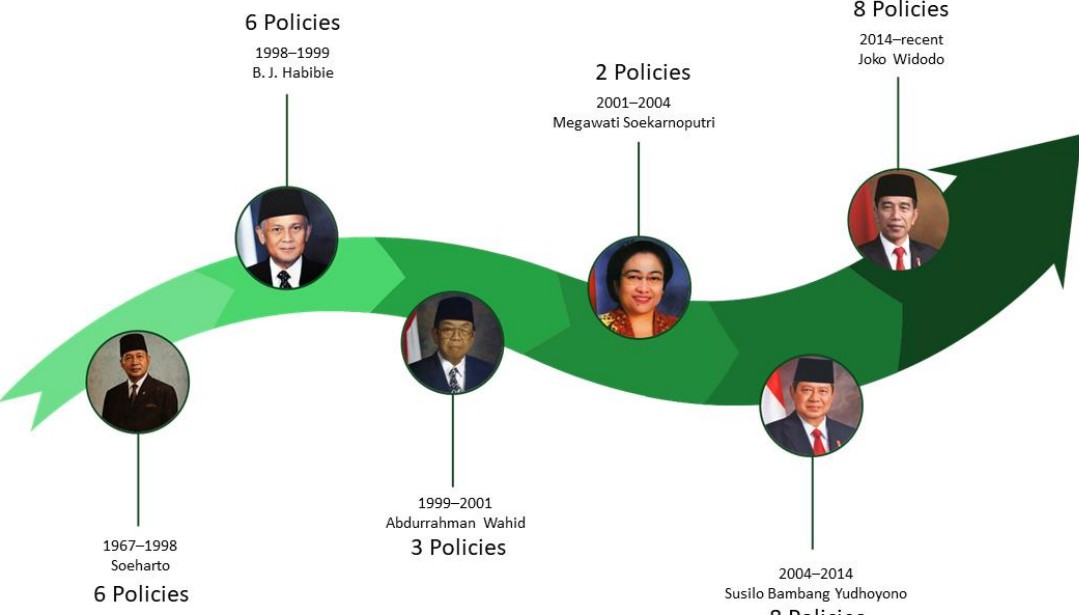

**Figure 8.** Number of policies based on Indonesia's presidential terms.

In Figure 8, we can observe that six Presidents have lead Indonesia in the period of 1969–2022. First, President Soeharto served for 32 years from 1967–1998. During that time, President Soeharto issued as many as seven policies related to digital transformation, Keppres, and Perpres. These policies discuss tourism development policies, the year of Indonesian tourist visits, and the establishment of the Indonesian Telematics Coordination Team in terms of digital transformation. The Indonesian Telematics Coordination Team is represented by Minister Parpostel based on Keppres 30/1997.

Then, during the term of President B. J. Habibie, in 1998–1999, there was a policy in the form of Keppres Regarding digital transformation, namely, Keppres 186/1998. This policy is a change from the previous policy during President Soeharto's term, namely, with the addition of the Implementation Team and Secretariat. Furthermore, in 1999–2001, when led by President Abdurrahman Wahid, there were three policies regarding digital transformation and tourism development: Keppres and Inpres. Through Keppres 11/2000, the Tourism and Arts Development Agency was formed, which also contains the development of human resources and information systems for tourism and the arts.

The next period was President Megawati Soekarnopoetri, who served from 2001–2004 and issued two policies regarding digital transformation. At that time, Inpres 3/2003 was issued, the initial policy of implementing e-government in Indonesia [57]. E-Government was carried out in four stages: preparation, maturation, stabilization, and utilization. Then, during the term of President Susilo Bambang Yudhoyono, which lasted from 2004–2014, there were eight policies in the form of Perpres, Keppres, and Inpres regarding digital transformation and tourism development. UU 11/2008 on digital information and transactions was established at that time. The policy is an effort to increase legal certainty for users and operators of information technology. In addition, Perpres 64/2014 in the form of a Tourism Coordination Team, which aims to improve the implementation of tourism at the level of policies, programs, and activities.

Finally, during the term of President Joko Widodo, which in this study was limited to only use data available until the end of 2022, there were as many as eight policies in the form of Perpres and Inpres, regarding digital transformation and tourism development in Indonesia. At that time, Indonesia's One Data (One Indonesian Data) policy was established as an integrated national data integration and the establishment of STP2D at the provincial and district/city levels. In the tourism development policy, there is a policy of electronically integrated business licensing services in the tourism sector. Based on this policy, Online Single Submission (OSS) was issued on behalf of the government to tourism businesses began to be implemented.

4.3.2. Tourism Information Center (TIC) and Tourism Information System (TIS)

The use of technology in tourism development is listed in various policies and programs. One of them is using technology through the Tourism Information Center (TIC) and Tourism Information System (TIS). TIC is listed in four policies, namely, Permenpar 1/2014, PermenLHK 13/2020, Permenparekraf/Baparekraf 3/2021, and Permenparekraf/Baparekraf 4/2022. TIC is in a strategic location, easy to see, and easily accessible to visitors with the support of facilities for people with disabilities. TIC itself is equipped with an electronic information display system and internet access.

Meanwhile, TIS is contained in two policies, Permenparekraf/Baparekraf 8/2020 and Permenparekraf/Baparekraf 4/2022. TIS is an application or software that can be used offline or online in at least two languages: English and Indonesian. The TIS can provide the most complete and up-to-date information about tourist destinations/areas. The content in TIS includes accommodations, the rental of motor vehicles and bicycles, tour/travel agents, money changers, hospitals, restaurants, markets/supermarkets, transport, calendar of events, tourist destinations, and *attraction* maps.

### 4.3.3. ICT Infrastructure Development in Tourism Areas

In technological transformation, adequate infrastructure is needed to support the process. In the criteria for tourism management in PermenLHK 76/2015 and PermenKP 14/2016, the development and availability of telecommunication facilities are proposed, namely, national cable or wireless provider networks. Then, based on KepDPD 16/2020, the government can provide fiscal and non-fiscal facilities to accelerate tourism development, one of which is tourism businesses that research and develop tourism technology and products.

## 5. Discussion

This section discusses two aspects, namely, trends in digital transformation policies and tourism development policies in Indonesia, as well as policy implications in the future.

### 5.1. Policy Trends for Digital Transformation and Tourism Development

In the internet age, ICT has been used more widely in tourism, making digital transformation occur [16]. This condition encourages the government to provide policies that regulate these two issues. Both types of policies are multisectoral policies that concern various actors [24], collaboration in the formulation, implementation, and including the evaluation of this policy becomes very important. This is reflected in the policies found to show that not only institutions that regulate tourism and communication technology issued policies related to digital transformation and tourism development, but other insitutions have also issued policies. Therefore, these two types of policies intersect with various cross-sector actors, such as the environment, marine and fisheries, the utilization of the state apparatus, and bureaucratic reform. In the pre-pandemic period, digital transformation and tourism development policies in Indonesia were no more than five policies (see Figure 3), and there were even years when no policy was set regarding these two things.

During the COVID-19 period in Indonesia, 2021 was the year when the most policies regarding digital transformation were established, namely, eight policies regarding SPBE and five tourism development policies. With this reflection, the United Nations considers the COVID-19 pandemic as a catalyst in accelerating digital development and transformation for e-learning/work, e-commerce, tourism, and travel [10,18]. In 1996, a year before the first digital transformation policy was issued based on this research, PP 67/1996 on tourism implementation already contained the use of electronic media or communication media for disseminating information about tourism businesses and other related information. The tourism sector has become more significant through the early stages of digital transformation compared to the previous period without the COVID-19 pandemic [58]. With the crisis caused by the COVID-19 pandemic, the government is better prepared by having an emergency plan for tourism for future possibilities [10].

Meanwhile, in other countries, such as Austria, the COVID-19 pandemic increased interaction between consumers and posts about tourist destinations by 133% likes, 50% comments, and 100% shares during the lockdown [59]. The study supports the social media presence of tourist destinations during the crisis. In addition, long before the COVID-19 pandemic, European Union countries had experienced a crisis since 2013, but managed to recover due to technological developments as the main catalyst [60]. If not handled properly, crisis handling will impact the reputation of policymakers and other actors, and, in a global crisis, such as the COVID-19 pandemic, it requires more intense cooperation and adaptation from multiple actors [61]

The condition of the COVID-19 pandemic, which has an impact on the tourism sector, on the other hand, is seen as an opportunity to transform while opening up new opportunities to develop sustainable tourism that is beneficial to social, economic, and environmental sectors [62]. In Indonesia, when the COVID-19 pandemic was happening in 2021, the government updated policies regarding guidelines for sustainable tourism destinations through Permenparekraf/Baparekraf 9/2021. This is aimed to reduce the potential spread of COVID-19 in the tourism sector. However, in this study, there are only three policies for

sustainable tourism in Indonesia until 2022. Meanwhile in Krabi, Thailand, stakeholders are working to develop the Five-Year and One-Year Krabi Tourism Management Action Plans as green policies and sustainable tourism management plans [15].

### 5.2. Policy Implications

Policy implications are examined by looking at digital transformation policy trends, which, during the COVID-19 pandemic, are proliferating, which implies that, in the following years, there will be various other policies that are derivatives of existing policies. This also happened when the world was attacked by Middle East Respiratory Syndrome (MERS), which triggered policies and affected tourism demand [63]. The policy is detailed because it is the operational basis for the use of technology. This shows that tourism development cannot be separated from the use of technology that allows opportunities for digital-based tourism policies to be born in Indonesia that regulate the use of technology in tourism management.

Looking at policy trends regarding digital transformation, which during the COVID-19 pandemic is proliferating, implies that in the following years, there will be various other policies that are derivatives of existing policies. In addition, with the policy trends found, digital transformation supports tourism development so that it will continue to intersect. The words "electronic", "digital", and "technological" in tourism development policies from 2004 to 2022 continue to be found. That statement shows that tourism development cannot be separated from the use of technology that allows opportunities for digital-based tourism policies to be born in Indonesia that regulate the use of technology in tourism management.

Meanwhile, the theoretical implication of this study is through the use of content analysis methods that are specific to policy content and are rarely used in legal documents [47]. This content analysis is used to analyze digital transformation policy and tourism policy in Indonesia's historical linkage and future implications. The results of this study can be utilized for further research and policy development for fields related to technology and tourism. The policy can be a guideline in digital-based tourism and evaluate the content of existing policies.

### 5.3. Limitations and Future Research Directions

The limitation of this study is that it only covers the interlinkage of digital transformation policies and tourism development nationally in Indonesia from 1879–2022. With that, further research can be carried out with a wider time frame and focus on digital transformation policies and tourism development toward post-COVID-19 pandemic recovery. In addition, data collection is carried out only through Indonesian and Dutch with JDIH sources belonging to the Indonesian central government so that further research can be developed into other languages, such as English, and with different sources, such as other public policy repositories during colonial occupations in Indonesia. Lastly, the issues of sustainability and sustainable tourism have not been so strong in this study. Therefore, there is a need for research that discusses sustainability policies in tourism.

**Author Contributions:** Conceptualization, W.D.N., I.W. and A.Z.M.; methodology: W.D.N., I.W. and A.Z.M.; software: W.D.N.; validation, W.D.N. and I.W.; formal analysis, W.D.N., I.W. and A.Z.M.; investigation, W.D.N. and I.W.; resources, W.D.N. and I.W.; data curation, W.D.N., I.W. and A.Z.M.; writing—original draft preparation, W.D.N.; writing—review and editing, W.D.N., I.W. and A.Z.M.; visualization, W.D.N. and A.Z.M.; supervision, I.W.; project administration, W.D.N. and I.W.; funding acquisition, I.W. All authors have read and agreed to the published version of the manuscript.

**Funding:** The APC was funded by Universitas Padjadjaran.

**Institutional Review Board Statement:** Not applicable.

**Informed Consent Statement:** Not applicable.

**Data Availability Statement:** Not applicable.

**Acknowledgments:** The authors wish to thank the Directorate of Research and Community Services Universitas Padjadjaran. This research is part of the Academic Leadership Grant research scheme, funded by Universitas Padjadjaran (2023–2026). The research is led by Ida Widianingsih.

**Conflicts of Interest:** The authors declare no conflict of interest.

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
