# Peer review of "The Linkage of Digital Transformation and Tourism Development Policies in Indonesia from 1879–2022: Trends and Implications for the Future"

_sustainability, doi:10.3390/su151310201_

Round 1

Reviewer 1 Report

I agree with the publication of the manuscript. However, authors should review the order of figures, there are two "Figures 1"; this should also be reflected in the text.

Author Response

Dear Reviewer 1,

Thank you very much for reviewing our article titled "The Linkage of Digital Transformation and Tourism Development Policies in Indonesia from 1879-2022: Trends and Implications for the Future".  We genuinely appreciate your insightful comments and suggestions, which have improved the quality of our work. Below, we address each of your points in detail:

Point 1: I agree with the publication of the manuscript. However, authors should review the order of figures, there are two "Figures 1"; this should also be reflected in the text.
Response 1:  We have changed by placing Figure 2 directly after Figure 1 because the explanations from the two figures are interrelated. The description is added under Figure 2.

We also attached the file, and please see the attachment. Once more, we appreciate your time and effort in reviewing our article. We hope that our responses adequately address your concerns and that the revised version meets the journal's standards.

Sincerely,
Windi Dwi Nanda
Corresponding Author
Universitas Padjadjaran
Jalan Ir. Soekarno Km. 21 Jatinangor, Sumedang 45363

Reviewer 2 Report

Dear Authors, thank you for the opportunity to review your manuscript.

It is very interesting and timely as it covers an aspect that is often underinvestigated and with a sound methodology, too.

Moreover, the relevance and ptiential impact in terms of real implications is important to attract the readership of the  journal.

However, please note that there are two minor and one key element to be addressed:

1) please remember that in the end of the introdcuction you should usually describe the structure of the article regarding each of the subsequent sections;

2) in the end, you refer to future implication: as the article focuses on policy aspects please remember that it would be better to give an immediate understanding to the reader about the content of each paragraph. Suggestion: change title to Policy impplications;

3) Finally, please note tha the discussion should be enriched by comparing your results with literature. For this reason, a good source of very recent changes in policy due to covid-19 is this one: https://www.sciencedirect.com/science/article/pii/S0278431922000093?via%3Dihub

Author Response

Dear Reviewer 2,

Thank you very much for reviewing our article titled "The Linkage of Digital Transformation and Tourism Development Policies in Indonesia from 1879-2022: Trends and Implications for the Future".  We genuinely appreciate your insightful comments and suggestions, which have improved the quality of our work. Below, we address each of your points in detail:

Point 1: Please remember that in the end of the introduction you should usually describe the structure of the article regarding each of the subsequent sections.

Response 1: We have added the article's description and structure about each section in the last paragraph of the Introduction section.

Point 2: in the end, you refer to future implication: as the article focuses on policy aspects please remember that it would be better to give an immediate understanding to the reader about the content of each paragraph. Suggestion: change title to Policy implications.

Response 2: We already changed the title of the “Future Implication” section to the “Policy Implications” section. This is based on the explanation given in the paragraph focusing on the implications for digital transformation policies and tourism development.

Point 3: Finally, please note tha the discussion should be enriched by comparing your results with literature. For this reason, a good source of very recent changes in policy due to covid-19 is this one: https://www.sciencedirect.com/science/article/pii/S0278431922000093?via%3Dihub

Response 3: We have added an explanation from the "Discussion" section by comparing the results obtained with the research conducted in other countries. And has included the article "Short-term rental market crisis management during the COVID-19 pandemic: Stakeholders' perspectives" to enrich this section.

We also attached the file, and please see the attachment. Once more, we appreciate your time and effort in reviewing our article. We hope that our responses adequately address your concerns and that the revised version meets the journal's standards.

Sincerely,
Windi Dwi Nanda
Corresponding Author
Universitas Padjadjaran
Jalan Ir. Soekarno Km. 21 Jatinangor, Sumedang 45363

Reviewer 3 Report

Thank you for giving this opportunity to review the manuscript entitled, “The linkage of digital transformation and tourism development policies in Indonesia from 1879-2022: Trends and implications for the future.

This manuscript provides rare information about policies from 1879 to 2022 in Indonesia.

The manuscript present the visualized outcomes of the NVivo as it is. Therefore, the audiences who did not code the rich materials, may not easily understand the contents of the analyzed policies.

Please provide more rich information about the policies of tourism development and digital transformation.

1. I recommend to revise the English writing

“wanted to see”

2. research questions

The research questions are not clearly presented.

What are the research questions on page 3?

3. problem statement

The problem statement should be presented in introduction.

4. literature review

The literature review should be written. Moreover, if the research used a content analysis, the research article need to present the theoretical framework and a guiding theory to analyze the contents of materials. In introduction, this manuscript describe the situations of the COVID-19 pandemic. The literature review is not well given.

5. method

In method, the coding process and ways to increase validity and reliability should be mentioned.

6. Figure 4

The content analysis may need to provide the meaning of the accumulated materials.

The changed name of the ministry in Indonesia does not provide meaning information.

7. Figures 5, 6, and 7

I do not think the figures 5, 6 and 7 present the results of the content analysis either quantitatively or qualitatively.

They are showing the nodes and the connects the authors coded via NVivo.

The authors should provide the readable and meaningful information in tables...or showing examples of the specific policies.

8. discussion

Discussion repeatedly presents the results. The specific information of policies is not well discussed. Because of the lack of theory-based information, this study do not present rich theoretical implications. Discussion also needs to provide practical implications.

9. limitations and future research directions

This section seems to be revised.

Author Response

Dear Reviewer 3,

Thank you very much for reviewing our article titled "The Linkage of Digital Transformation and Tourism Development Policies in Indonesia from 1879-2022: Trends and Implications for the Future".  We genuinely appreciate your insightful comments and suggestions, which have improved the quality of our work. Below, we address each of your points in detail:

Point 1: I recommend to revise the English writing “wanted to see”

Response 1: We have made changed to “analyze”.

Point 2: Research questions. The research questions are not clearly presented. What are the research questions on page 3?

Response 2: We already changed the title of the “Future Implication” section to the “Policy Implications” section. This is based on the explanation given in the paragraph focusing on the implications for digital transformation policies and tourism development.

Point 3: Problem statement. The problem statement should be presented in introduction.

Response 3: We have added a sentence of the problem statement in the second last paragraph in the Introduction section.

Point 4: Literature review. The literature review should be written. Moreover, if the research used a content analysis, the research article need to present the theoretical framework and a guiding theory to analyze the contents of materials. In introduction, this manuscript describe the situations of the COVID-19 pandemic. The literature review is not well given.

Response 4: We have added the Theoretical Background section after the Introduction section. The section described concept digital transformation, tourism, and their policy.

Point 5: Method. In method, the coding process and ways to increase validity and reliability should be mentioned.

Response 5: We already added the statement about the coding process when operating Nvivo 12 Pro. And background and ways to increase validity and reliability in the Method section.

Point 6:  Figure 4. The content analysis may need to provide the meaning of the accumulated materials. The changed name of the ministry in Indonesia does not provide meaning information.

Response 6: We have added an explanation from the Figure 4 to provide meaningful information and enriched the result. The description explained the characteristic and the supported policies in changes of each nomenclature minister in charge of tourism affairs in Indonesia.

Point 7:  Figures 5, 6, and . I do not think the figures 5, 6 and 7 present the results of the content analysis either quantitatively or qualitatively. They are showing the nodes and the connects the authors coded via NVivo. The authors should provide the readable and meaningful information in tables...or showing examples of the specific policies.

Response 7: We have changed the content analysis o Figure 5 to Table 2 because the project map is not as readable as Figure 6 and Figure 7 (now Figure 5 and Figure 6). And we have added examples of specific policies that set about sustainable tourism because it follows the phenomenon and the background of the research.

Point 8: Discussion. Discussion repeatedly presents the results. The specific information of policies is not well discussed. Because of the lack of theory-based information, this study do not present rich theoretical implications. Discussion also needs to provide practical implications.

Response 8: We changed the focus of the Discussion to elaborate on the result with other research and literature review. The Discussion also compared the findings to other research.

Point 9: Limitations and future research directions. This section seems to be revised.

Response 9: We have revised the limitations and changed the statement about the future research direction.

We also attached the file, and please see the attachment. Once more, we appreciate your time and effort in reviewing our article. We hope that our responses adequately address your concerns and that the revised version meets the journal's standards.

Sincerely,
Windi Dwi Nanda
Corresponding Author
Universitas Padjadjaran
Jalan Ir. Soekarno Km. 21 Jatinangor, Sumedang 45363

Reviewer 4 Report

The methodology is solid, based on content analysis. The tool used is NVivo, one of the best for qualitative data analysis. The theme of sustainability and sustainable tourism is touched, but for me it is not clear how relevant this theme emerged from the content analysis. May the authors describe better what patterns they found in their data about sustainability? Is there any link between sustainability and digital transformation? May the authors clarify in the discussion how sustainability is present or not as theme in their data? if sustainability emerged, but less than other themes, this can be presented and argumented, with evidences from data, in the discussion and In future research. 

Author Response

Dear Reviewer 4,

Thank you very much for taking the time to review our article titled "The Linkage of Digital Transformation and Tourism Development Policies in Indonesia from 1879-2022: Trends and Implications for the Future".  We truly appreciate your insightful comments and suggestions, which have improved the quality of our work. Below, we address each of your points in detail:

Point 1: The theme of sustainability and sustainable tourism is touched, but for me it is not clear how relevant this theme emerged from the content analysis. May the authors describe better what patterns they found in their data about sustainability? Is there any link between sustainability and digital transformation? May the authors clarify in the discussion how sustainability is present or not as theme in their data? if sustainability emerged, but less than other themes, this can be presented and argumented, with evidences from data, in the discussion and In future research.

Response 1: We added the theme about sustainability and sustainable tourism on the Theoretical Background, Result in subsection 4.2.1., Discussion, Limitations and Future Research Directions. And we added two policies about sustainable tourism in Indonesia. Theoretical Background and Discussion describe the relationship between sustainability and digital transformation. The clarification of sustainability is presented in Discussion and explains the presence of sustainability policy but less than other topics based on the finding.

We also attached the file, please see the attachment. Once again, we appreciate your time and effort in reviewing our article. We hope that our responses adequately address your concerns and that the revised version meets the standards of the journal.

Sincerely,

Windi Dwi Nanda

Universitas Padjadjaran

Jalan Ir. Soekarno Km. 21 Jatinangor, Sumedang 45363

Round 2

Reviewer 3 Report

Thank you for your revision based on the comments. 

I found important information in the contents. However, the table 2, Figures 6-7 do not present readable results for readers. These table and figures should be reconsidered. 

Author Response

Dear Reviewer 3,

We appreciate the time you dedicated to reviewing our paper titled "The Linkage of Digital Transformation and Tourism Development Policies in Indonesia from 1879-2022: Trends and Implications for the Future". Your insightful comments and suggestions have been invaluable in enhancing the quality of our research. In response to your feedback, we have implemented the following revisions:

Point : I found important information in the contents. However, the table 2, Figures 6-7 do not present readable results for readers. These table and figures should be reconsidered. 

Response : We have added details to the explanations for Table 2, Figures 6-7. These explanations are based a specific explanation of these policies and the basis for the argumentation of policy classification on the topics set. Therefore, the visualization of the table and figures can be interpreted by reading the descriptions (under Table 2, Figures 6-7).

We also attached the file, please see the attachment We would like to express our gratitude once again for your time and effort in reviewing our manuscript. We believe that the revisions we have made effectively address your concerns and significantly enhance our research quality.

Sincerely,

Windi Dwi Nanda

Universitas Padjadjaran

Jalan Ir. Soekarno Km. 21 Jatinangor, Sumedang 45363
